# Regional variations in *Helicobacter pylori* infection, gastric atrophy and gastric cancer risk: The ENIGMA study in Chile

Rolando Herrero[1,2]*, Katy Heise[3], Johanna Acevedo[4], Paz Cook[5,6], Claudia Gonzalez[7], Jocelyne Gahona[8], Raimundo Cortés[8], Luis Collado[9], María Enriqueta Beltrán[3], Marcos Cikutovic[8], Paula Gonzalez[2†], Raul Murillo[1], Marcis Leja[10], Francis Megraud[11,12], Maria de la Luz Hernandez[1], Sylvaine Barbier[1], Jin Young Park[1], Catterina Ferreccio[5,6]*, for the ENIGMA Chile study group¶

1 International Agency for Research on Cancer, Lyon, France, 2 Agencia Costarricense de Investigaciones Biomédicas, Fundación INCIENSA, Costa Rica, 3 Hospital Cancer Registry, Hospital Base Valdivia, Valdivia, Chile, 4 Epidemiology Department, Health Ministry, Santiago, Chile, 5 Public Health Department, School of Medicine, Pontificia Universidad Católica de Chile, Santiago, Chile, 6 Advanced Center for Chronic Diseases, ACCDis, Santiago, Chile, 7 Secretaría Regional Ministerial de Salud, Antofagasta Region, Chile, 8 Faculty of Health Sciences, Universidad de Antofagasta, Antofagasta, Chile, 9 Institute of Biochemistry and Microbiology, Faculty of Sciences, Universidad Austral de Chile, Valdivia, Chile, 10 Institute of Clinical and Preventive Medicine, University of Latvia, Riga, Latvia, 11 Bordeaux University, INSERM, UMR1053, BaRITOn, Team 2 "Helicobacter Infection, Inflammation and Cancer", Bordeaux, France, 12 Pellegrin University Hospital, Bacteriology Laboratory, French National Reference Center for Campylobacters and Helicobacters, Bordeaux, France

† Deceased.
¶ The complete membership of the author group can be found in the acknowledgments.
* rherrero@acibcr.com (RH); catferre@gmail.com (CF)

## Abstract

### Background

Regional variations in gastric cancer incidence are not explained by prevalence of *Helicobacter pylori*, the main cause of the disease, with several areas presenting high *H. pylori* prevalence but low gastric cancer incidence. The IARC worldwide *H. pylori* prevalence surveys (ENIGMA) aim at systematically describing age and sex-specific prevalence of *H. pylori* infection around the world and generating hypotheses to explain regional variations in gastric cancer risk.

### Methods

We selected age- and sex-stratified population samples in two areas with different gastric cancer incidence and mortality in Chile: Antofagasta (lower rate) and Valdivia (higher rate). Participants were 1–69 years old and provided interviews and blood for anti-*H. pylori* antibodies (IgG, VacA, CagA, others) and atrophy biomarkers (pepsinogens).

### Results

*H. pylori* seroprevalence (Age-standardized to world population) and antibodies against CagA and VacA were similar in both sites. *H. pylori* seroprevalence was 20% among

**Funding:** The work in Chile was supported in part by a grant from Fondo Nacional de Investigación y Desarrollo en Salud (FONIS grant SA1312007) and the pepsinogen work in Latvia by a grant from the Latvian Council of Science (nr. lzp-2018/1-0135).

**Competing interests:** The authors have declared that no competing interests exist.

children <10 years old, 40% among 10–19 year olds, 60% in the 20–29 year olds and close to or above 80% in those 30+ years. The comparison of the prevalence of known and potential *H. pylori* cofactors in gastric carcinogenesis between the high and the low risk area showed that consumption of chili products was significantly higher in Valdivia and daily non-green vegetable consumption was more common in Antofagasta. Pepsinogen levels suggestive of gastric atrophy were significantly more common and occurred at earlier ages in Valdivia, the higher risk area. In a multivariate model combining both study sites, age, chili consumption and CagA were the main risk factors for gastric atrophy.

## Conclusions

The prevalence of *H. pylori* infection and its virulence factors was similar in the high and the low risk area, but atrophy was more common and occurred at younger ages in the higher risk area. Dietary factors could partly explain higher rates of atrophy and gastric cancer in Valdivia.

## Impact

The ENIGMA study in Chile contributes to better understanding regional variations in gastric cancer incidence and provides essential information for public health interventions.

## Introduction

Gastric cancer (GC) causes almost 800,000 yearly deaths worldwide, and despite declining trends, disease burden will not decline for decades because of population growth and aging [1]. GC exhibits extreme regional variation, with reported age-standardized incidence rates (ASIR) for males ranging from 1.4/100,000 in Eastern Cape, South Africa to 144/100,000 in Yanting, China [2]. In Iran, GC incidence varies 5-fold between the Northern and the South-Eastern region (50 and 10 per 100,000 men respectively) [3]. In Chile, cancer registries report a consistent pattern over several years of higher ASIR of GC in Valdivia (33.1 and 13.1 per 100,000 men and women, respectively) compared with Antofagasta (21.2 and 7.8) in 2008–2010, with corresponding mortality rates in 2009 about twice as high in Valdivia as in Antofagasta (31.2 in Valdivia vs. 10.9 in Antofagasta in both sexes) [2, 4].

The main risk factor for GC is chronic *Helicobacter pylori* (*H. pylori*) infection, present in an estimated 50% of the world population [5]. Infection is usually acquired in childhood and generally persists without symptoms for life. *H. pylori* causes chronic gastritis, which can lead to peptic ulcer disease, but also to premalignant lesions (atrophic gastritis, intestinal metaplasia, dysplasia) and finally GC in the series of events known as the Correa cascade [6]. The bacterium is an International Agency for Research on Cancer (IARC) Group 1 carcinogen [5] and the attributable fraction for non-cardia GC is close to 90% [7]. However, it is clear that GC etiology is multifactorial, with bacterial, environmental and genetic factors also playing a not fully clarified role.

Prevalence of *H. pylori* is declining in most regions, particularly in younger generations [8–10], most likely explaining observed reductions in GC rates. However, limited data exist on the age and sex-specific prevalence of *H. pylori* infection in general populations, particularly in low- and middle-income countries. Many previous studies included symptomatic patient

series and used variable laboratory methodologies, but standardised information on population-based samples is essential to plan public health interventions for GC control.

Considering the etiologic link between *H. pylori* and GC, it would be expected that regional variations in GC incidence would correlate closely with prevalence of *H. pylori*, as described for other infection-related cancers (e.g., HPV and cervical cancer [11]). In areas where *H. pylori* prevalence is low, GC is rare, but many areas with high *H. pylori* prevalence have low GC rates, as observed in some African countries. In Uganda, prevalence of *H. pylori* has been reported around 90% [12] while GC incidence is low, with ASIR of 5.9/100,000 among men [2]. Similar lack of association is reported among African descendants in Colombia [13], and has also been described in areas with uniform ethnic backgrounds like Chile, Mexico, Costa Rica [14] or Iran [3]. Regional differences are likely related to host genetics, bacterial factors (related to *H. pylori* or gastric microbiota), environmental factors (e.g. dietary habits, sodium intake, tobacco use, altitude) or unidentified exposures.

Gastric atrophy is recognized as a critical step in the Correa pathway to intestinal-type gastric cancer, with consistent associations with gastric cancer, and it can be used as a surrogate of gastric cancer risk in population-based studies [15]. Risk factors for atrophy have been investigated in several studies, including age, family history, education, male sex, *H. pylori* positivity, CagA and VacA and dietary factors such as consumption of coffee, spicy and salty food [16–19].

The ENIGMA (Epidemiological iNvestigatIon of Gastric MAlignancies) studies are a series of global prevalence surveys coordinated by IARC in high- and low-risk areas of GC. We are conducting population-based studies with standardized questionnaire data, biological specimens and laboratory procedures to describe worldwide epidemiology of *H. pylori*, predict future GC trends and generate hypotheses explaining regional and ethnic differences in risk. In this manuscript we present the results of the first ENIGMA study, conducted in two areas in Chile with different risk of GC.

## Materials and methods

### Study population

We selected two regions in Chile with different rates of GC mortality: 31.2 in Valdivia vs. 10.9 in Antofagasta in both sexes. Antofagasta is in the North with dry weather and more affluent economy and Valdivia is in the South with a rainy climate and less affluent population.

According to the ENIGMA protocol, in each area, age- and sex-stratified population-based samples were selected. The reference population were residents of the cities of Antofagasta and Valdivia aged 1 to 69 years old (25 men and 25 women in each 5-year age group). In each city, groups of socio-economically homogeneous blocks were defined, in numbers proportional to the total of blocks in each conglomerate. In a first phase, blocks were randomly selected within each conglomerate. In a second phase, systematic selection of households was carried out within selected blocks, initiating the count at the South-East corner of each block and contacting every third house. In a third phase individuals were chosen within households according to eligibility (see below).

In Antofagasta, the sampling frame was the entire city while in Valdivia it was the Barrios Bajos Sector, representing the city's socio-economic diversity. Recruitment staff and interviewers were trained by Catholic University team and IARC researchers. In Antofagasta, recruitment was performed by advanced students from the Health Sciences Faculty of the University of Antofagasta, supported by educators in Obstetrics, Medical technologies and Nutrition. In Valdivia, nurse aides with population survey experience carried out recruitment. Recruitment

period was from 10 May 2014 to 11 August 2015. Refusers were replaced until the required sample size of ~700 subjects in each site was obtained.

The study was approved by Ethics Committee of IARC (IEC No.14-17), the Ethics and Scientific Committe of the Faculty of Medicine of Pontificia Universidad Católica de Chile. [Comité Etico Científico de la Facultad de Medicina, Santiago, Chile] and the Ethics and Scientific Committe of the Comquimbo Health Service Direction, La Serena Chile [Comité Etico Científico de la Dirección de Servicio de Salud Coquimbo (La Serena, Chile)].

Eligible adults signed informed consent. Children between 12 and 18 provided informed assent and parental informed consent, and for children under 12 only parental informed consent was obtained.

Eligibility criteria included mental and physical competence and no history of GC. All procedures were conducted at home. A questionnaire was administered including socio-demographic and occupational information, educational level, smoking, medical history, medication use, diet (food frequency), alcohol consumption, exposure to pesticides in addition to standardized anthropometric measurements. Blood (20 ml) was collected and serum, plasma and buffy coats aliquots produced. Urine and faecal samples were also collected for future studies.

## Laboratory analyses

Anti-*H. pylori* IgG class antibodies were measured at IARC with an ELISA test according to the manufacturer's instructions, based on an enzyme immunoassay technique with partially purified *H. pylori* bacterial antigen adsorbed on a microplate and a detection antibody labelled with horseradish peroxidase (Biohit Plc, Helsinki, Finland).

Among participants over 40 years (n = 616), we conducted additional analyses of *H. pylori* and its virulence factors and tested for pepsinogens (Pg) I and II as gastric atrophy markers (see below). This age group was selected as the most likely to be linked to current GC incidence rates.

The Helicoblot 2.1 immunoblot kit (Genelabs Diagnostics, Singapore) was performed at Bordeaux University, France to detect IgG antibodies against specific proteins of *H. pylori*, including the product of the cytotoxin-associated gene A (Cag A) and the vacuolating cytotoxin A (Vac A) according to manufacturer's recommendations. Helicoblot 2.1 uses a Western blot from bacterial lysate and includes recombinant antigen of *H. pylori* with high predictive value for detecting current *H. pylori* infection.

Pepsinogens I and II were blindly measured with a latex-agglutination test-system (Eiken Chemical, Tokyo, Japan) at University of Latvia. The cut-off value for pepsinogens considered to be associated with any gastric mucosal atrophy was PgI ≤70 ng/ml and PgI/PgII ≤3 while PgI ≤30 ng/ml and PgI/PgII ≤2 was considered marker of severe atrophy, according to manufacturer's reference values and previous validation by Dr Leja and collaborators [20].

## Statistical analyses

For all age groups, demographic characteristics and *H. pylori* seroprevalences were presented by site and compared using Chi-Square or Fisher tests. Age adjustment of estimates was carried out with the direct method based on Segi World Standard population [21] and also on the study population. Statistical significance of differences by study site of risk factors, *H. pylori* antibodies, virulence markers and pepsinogens was assessed with Chi-Square tests.

The factors associated with the Valdivia study site and atrophy were compared using univariable and multivariable logistic regression models to estimate odds ratios (ORs) and 95% CIs. These variables included *H. pylori* antibodies, gender, ethnicity, age, weight, smoking

status (current vs former or never), alcohol drinking habit (binge drinking, alcohol flush reaction), and education (≥8 vs <8 years). Dietary factors included daily intake of fruits, green vegetables, other vegetables, each and any type of chili, salted or smoked food and salt intake frequency. Environmental exposure, e.g. pesticides and medical history such as asthma, diabetes, tuberculosis, anaemia, colon inflammation, and drug use including anti-parasite drugs and antibiotics in the last 6 months were also tested in the univariable model. These factors were then included in the stepwise selection of the model based on their *a priori* clinical significance or *p*-values for the OR below a threshold of 0.2. The final model was validated by referring to the Akaike's information criterion and the Bayesian information criterion comparing the combination of the variables selected in the stepwise process [22]. All the statistical analyses were conducted using STATA (version 13, Stata Corporation, College Station, TX), using a significance level alpha of 0.05.

## Results

Fig 1A shows field work results in Antofagasta. Of 440 houses visited, 301 had at least one eligible subject (68.4%). A total of 923 potentially eligible subjects were identified in those households, of which 107 were not eligible. Of the total of 816 eligible subjects, 690 participated (participation rate: 84.5%). In Valdivia (Fig 1B) 724 houses were visited and 370 had at least one eligible subject (51.1%). There were 1194 potentially eligible subjects, of which 991 were eligible and 705 agreed to participate (71.1%).

Considering all ages, the sample was balanced by age group, but Antofagasta had proportionally more women than Valdivia (S1 Table). In Valdivia, there was a higher proportion of subjects of Mapuche origin while in Antofagasta there were more Aymara and more subjects of other ethnicities. Overweight and obesity, calculated only for subjects over 18, was high in both places, with approximately 75% overweight or obese.

Overall age-adjusted *H. pylori* ELISA seroprevalence was similar in both cities when considering ages 1-to70 (Table 1). Crude prevalence was 67% (95%CI = 63–71) in Antofagasta (lower GC risk area) and 63% (95% CI = 60–67) in Valdivia. Age-adjusted prevalence was 58% (95%

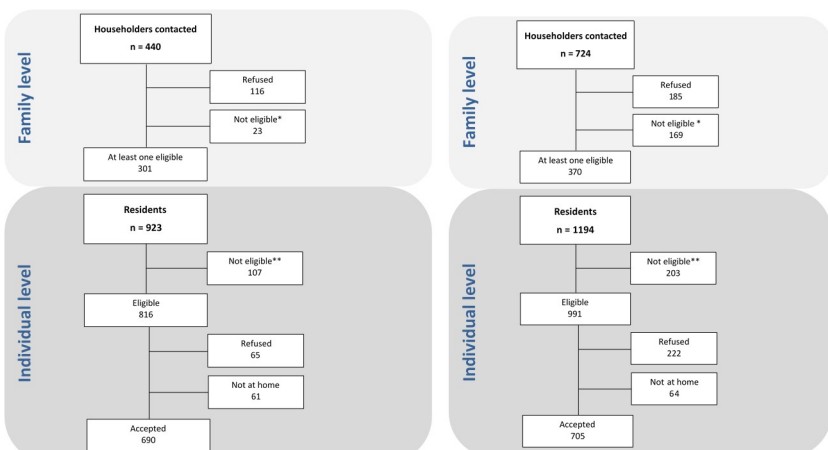

**Fig 1.** (a) Flow diagram of field work in Antofagasta. *Empty houses or no contact after several attempts. **Not eligible includes 37 subjects who did not fulfil the eligibility criteria and 70 subjects who were enumerated but not recruited because the required numbers for the corresponding categories of sex and age were already enrolled. (b) Flow diagram of field work in Valdivia. *Empty houses or no contact after several attempts. ** Not eligible includes 33 subjects who did not fulfil the eligibility criteria and 170 subjects who were enumerated but not recruited because the required numbers for the corresponding categories of sex and age were already enrolled.

**Table 1. *H. pylori* prevalence determined by ELISA by study site (all ages).**

| | | Study site | | | |
|---|---|---|---|---|---|
| | | Antofagasta | | Valdivia | |
| | | N = 690 | (95% CI) | N = 705 | (95% CI) |
| | | N (%) | | N (%) | |
| *H. pylori* status (crude) | | | | | |
| | Positive | 432 (67) | (63–71) | 434 (63) | (60–67) |
| | Negative | 211 (33) | (29–37) | 250 (37) | (33–40) |
| | Missing | 47 | | 21 | |
| *H. pylori* prevalence (adjusted)† | | 58 | [51–69] | 56 | [49–66] |
| | Women | 57 | [48–70] | 53 | [45–62] |
| | Men | 59 | [50–69] | 58 | [49–70] |

† Based on the Segi World Standard (Segi M, Kurihara M. Cancer mortality for selected sites in 24 Countries no. 6 (1966–1967) Nagoya, Japan Cancer Society; 1972).

CI = 51–69) in Antofagasta and 56% (95%CI = 49–66) in Valdivia. These estimates were also similar by gender with a statistically not significant higher prevalence in men than women in Valdivia (58% vs 53% in men and women, respectively, *p* = 0.19). Considering both sites combined, there were no clear differences in ELISA seroprevalence by gender among younger subjects, although prevalence was higher in men in all age groups over 30 years old (S1 Fig), with a *p* = 0.03 for all subjects over 30 combined.

In Antofagasta, but not in Valdivia, *H. pylori* prevalence was significantly higher in participants not born in the area compared with those born in Antofagasta (*p* = 0.007).

Seroprevalence of *H. pylori* was 20% among children under 10 years with a gradual increase thereafter (Fig 2). In Antofagasta, seroprevalence peaked at close to 90% in age group 50–59 with a subsequent decline to under 80% in subjects over 60. In Valdivia, seroprevalence peaked in the 40–49 age group with a subsequent decline to under 70% in those over 60. In the 50–59 age group, the difference between Antofagasta (prevalence = 90%, 95%CI = 83–95) and Valdivia (prevalence = 75%, 95%CI = 66–83) was statistically significant (*p* = 0.04).

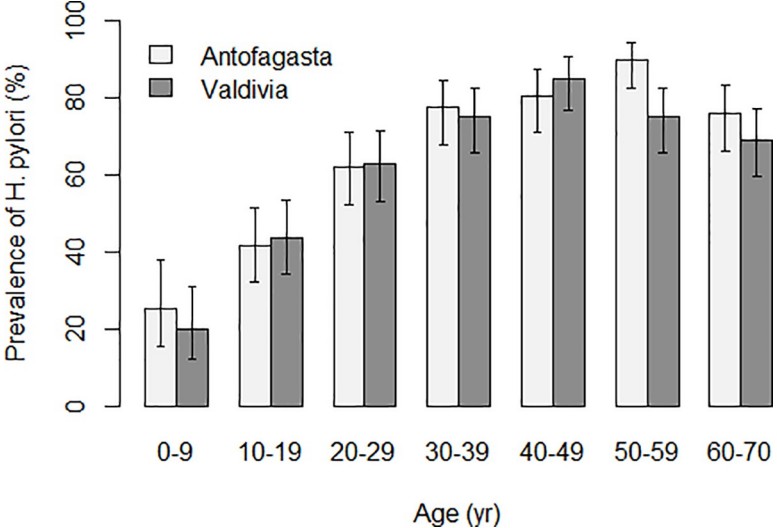

**Fig 2. *H. pylori* prevalence determined by ELISA by 10 years age group and site.**

S2 Table compares *H. pylori* detection by ELISA with results based on immunoblot methods in subjects over age 40, with similar results for the two methods in both cities. The prevalence of CagA and VacA antibodies was also comparable, with estimates very similar to those of overall *H. pylori* positivity. Antibodies against other *H. pylori* antigens (37 kDa, 35 kDa, 30 kDa and 19.5 kDa) had varying levels of positivity, but in all cases were very similar in the two areas. More than 90% of *H. pylori* positive subjects had antibodies against CagA and VacA in both sites, and positivity of antibodies against the other antigens among the *H. pylori* positives was again similar by site. Only a handful of the *H. pylori* negatives by ELISA were positive for these antigens. The minor difference in the results between ELISA and Helicoblot assays may be explained by the fact that the latter could be a better indicator of a current *H. pylori* infection [23].

Fig 3 presents a forest plot with a multivariate model of the differences in prevalence of risk factors between study sites (age over 40); factors more common in Valdivia are to the right of the unity, with Antofagasta to the left. There were more men recruited in Valdivia. Chilean Hispanic ethnicity, daily intake of chili and history of parasite drug use were more common in Valdivia, the higher risk area. Age between 50–59, alcohol flush reaction, daily intake of alcohol, daily consumption of non-green vegetables (e.g., cucumber, radish, pepper), adding salt to food at the table, use of antibiotics, and history of pesticide exposure were significantly more common in Antofagasta. The strongest OR associated with the Valdivia study site was intake of chili peppers with an OR of 8.8 (95%CI = 4.2–20.6). A history of anemia was more common in Antofagasta (26% vs 16%, *p* = 0.002) but it was not retained in the final model. Similarly,

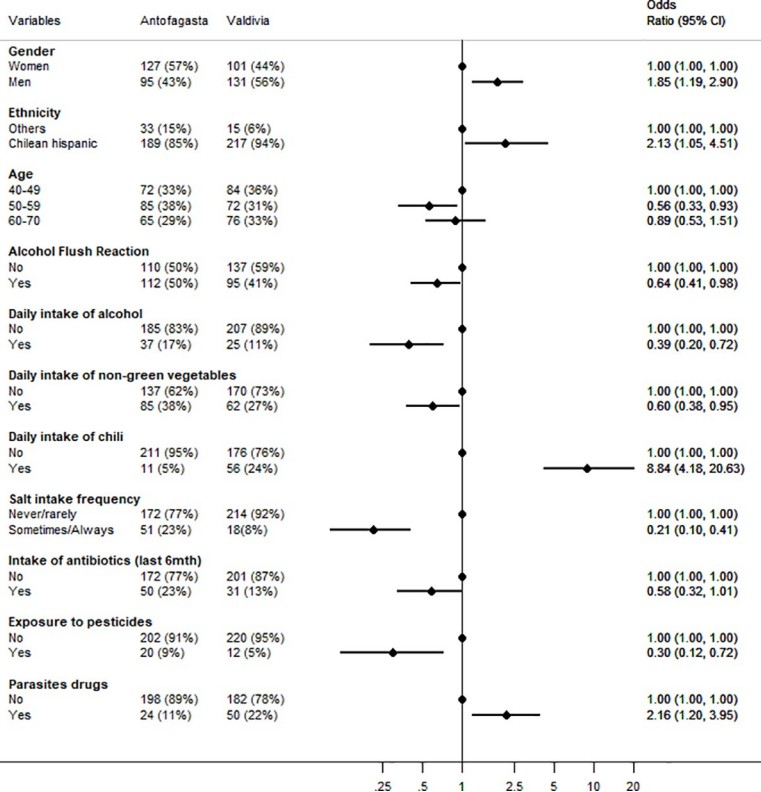

**Fig 3. Forest plot of factors associated with the Valdivia study site among participants aged 40 and over (multivariable logistic regression).** The first two columns present the prevalence for each site, the right part of the graph presents the adjusted odds ratio for that variable in the high-risk vs low risk site.

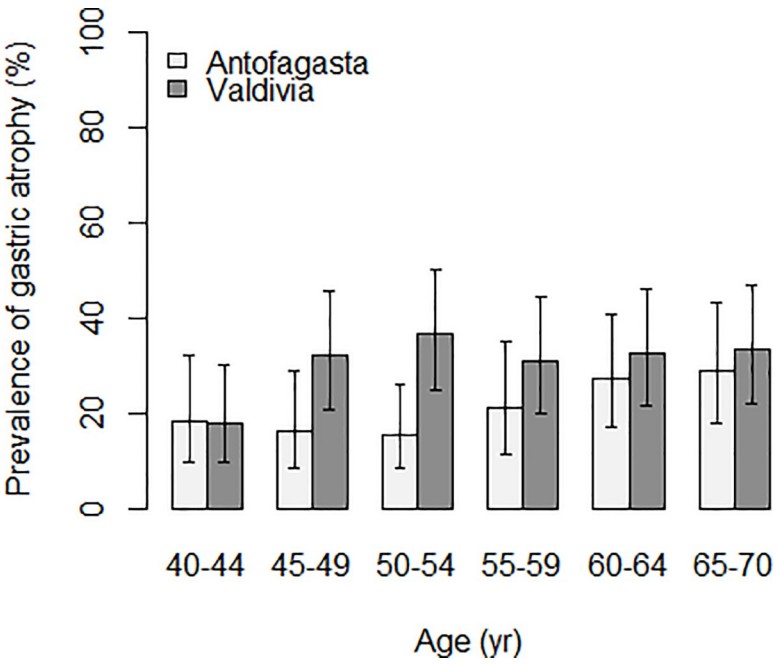

**Fig 4. Prevalence of atrophy (based on pepsinogen testing\*) by 5 years age group and study site among 40 years and older.** * Pepsinogens I and II were measured with a latex-agglutination test-system (Eiken Chemical, Tokyo, Japan) and cut off values for any gastric mucosal atrophy were PgI≤70 ng/ml and PgI/PgII≤3.

current smoking was more common in Valdivia (34% vs 25%), but it was not retained in the final model. Education, binge drinking, consumption of salted or smoked foods, fruits and green vegetables were not different between sites in this age group.

Prevalence of atrophy and severe atrophy, as determined by PgI and II levels, showed a significant difference between the two sites. Atrophy prevalence was significantly higher in the high risk region (31% vs 21%, $p = 0.005$), and this difference was stronger when restricted to *H. pylori* positive subjects by the Helicoblot method ($p < 0.001$). Advanced atrophy was 11% in Valdivia and 7% in Antofagasta ($p = 0.12$, data not in tables). Fig 4 presents the comparison of the prevalence of atrophy by age and site. Among subjects aged 40–44 years, atrophy was less than 20% in both sites. However, while in Antofagasta (the low-risk area), prevalence remained below 20% until age 54, in Valdivia it increased to more than 30% after age 45. The prevalence of atrophy in the age group of 50–54 was more than twice higher in Valdivia (36.5%) compared with Antofagasta (15.4%) and the difference was statistically significant ($P = 0.008$). In Antofagasta, prevalence increased slowly with age but did not reach 30% even in the older age group. There was no difference in the prevalence of atrophy according to gender in either site. The mean of the PgI/PgII ratios was significantly lower in Valdivia (3.6) compared to Antofagasta (4.0) ($p = 0.008$), and there were no differences by age and sex.

In a multivariate model of risk factors associated with atrophy (including severe atrophy) combining both sites, atrophy was positively associated with being from Valdivia, daily consumption of chili pepper and detection of antibodies against CagA while negatively associated with higher educational level (Fig 5). Smoking was not associated with risk of atrophy in this dataset.

## Discussion

This is the first of a series of IARC international prevalence surveys to clarify the epidemiology of *H. pylori* infection in high- and low-risk areas of GC, in an effort to predict GC incidence in

young cohorts and generate hypotheses to explain its regional variations. Etiologic clues might then lead to identification of biomarkers or preventive interventions.

*H. pylori* infection causes 90% of non-cardia GC, and ongoing reductions in incidence are likely a result of reductions in *H. pylori* prevalence, which has been documented in many areas, mainly in high-income countries [9]. However, comparable data on *H. pylori* prevalence are scarce, as the studies have used dissimilar methodologies and laboratory methods. In addition, GC does not correlate as expected with *H. pylori* prevalence in adults, with many areas having high prevalence of *H. pylori* but low GC incidence rates. The term 'African enigma' was coined by Holcombe in 1992 [24] to describe the discrepancy between prevalence of infection and clinical manifestations of *H. pylori* (peptic ulcer and GC). Several authors described similar enigmas in Asia and Latin America and proposed different explanations, including variation in *H. pylori* strains, virulence factors or host genetics [25, 26]. Graham *et al* dismissed the findings as medical myths, recommending to focus on underlying patterns of gastritis (non-atrophic vs atrophic) and environmental factors [27]. In this study, we analyse both bacterial and environmental factors in conjunction with an indicator of the presence of atrophic gastritis. We believe that the same factors causing gastritis are likely to explain the regional and individual differences in GC risk.

We observed a very similar seroprevalence of *H. pylori* in the two communities in Chile, with an age-adjusted estimate close to 60% considering all age groups. Prevalence increased clearly with age from 20% in children under 10 to a peak at ages 40 or 50 between 80% and 90% depending on the location. Notably, *H. pylori* prevalence peaked earlier in the higher risk area, and declined faster and to a lower level than in the low risk area, where prevalence peaked later and at a higher level and declined less than in the high risk area. *H. pylori* presence in the stomach decreases with advancing atrophic changes [28], and disappearance of the bacteria

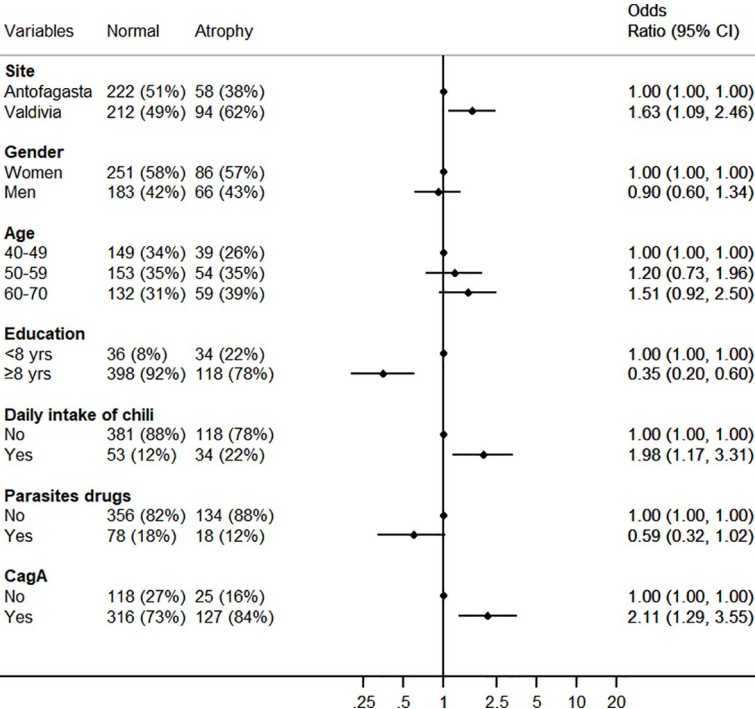

**Fig 5. Forest plot of the factors associated with atrophy among participants aged 40 and older (multivariable logistic regression).**

results after some time in serology becoming negative [29]. We hypothesize that in the higher risk area, the earlier occurrence of atrophy might be the explanation for the earlier peak in *H. pylori* infection and the rapid decline afterwards. In fact, we observed that serologic evidence of atrophy occurred earlier and was higher in Valdivia than in Antofagasta (see below).

Overall, *H. pylori* infection was as common in men as in women (both sites combined), but higher rates were detected in men over 30 years. Given that most infections initiate in childhood, and that 90% of GCs are associated with infection, the lower prevalence in younger generations likely predicts future declines in GC incidence in Chile, as has been observed elsewhere. We plan to work on a quantitative model to estimate future risk of GC in the countries participating in ENIGMA.

In the 40+ year age group, the most likely to explain current rates of GC, we compared prevalence of *H. pylori* by ELISA and Helicoblot 2.1, with an 85% agreement and comparable prevalence estimates. Seroprevalence of *H. pylori* was around 80% among subjects over 40 years with no differences by site. Antibodies against CagA, VacA or the other antigens explored were also comparable by site. CagA and VacA antibodies were almost universal among *H. pylori* positive subjects and very rare in *H. pylori* negatives. CagA and VacA are important *H. pylori* virulence factors and have been associated with risk of GC [30, 31]. In some studies, CagA was more common in high- than in low-risk areas [25, 27], but not in our study and others [32, 33].

Among the interesting findings regarding differences in risk factors between the two sites was consumption of non-green vegetables, which was more common in the low risk area, but there were no significant differences for consumption of other vegetables and fruits. In the Chilean National Health Survey of 2009–2010, consuming at least five portions of fruit and vegetables daily was 32.5% in Antofagasta compared to 1.4% in the Valdivia region, but daily portions of fruits and vegetables was not significantly different by region [34].

Consumption of chili products was more common in the high risk area. Chili has been proposed as a direct irritant of the gastric mucosa but studies have been inconsistent on potential benefits or harms of its active principle capsaicin [35, 36]. López-Carrillo and colleagues explored potential interactions among capsaicin, *H. pylori* virulent factors and genetic factors and showed that moderate to high capsaicin consumption was associated with increased risk of GC in genetically susceptible individuals (*IL1B-31C* allele carriers) infected with CagA positive strains [37].

Alternatively, chili could be contaminated with aflatoxins as has been demonstrated for some chili products in Chile [38]. Aflatoxin is a known group 1 carcinogen that causes cancer of the liver [39] and its dietary consumption (by questionnaire) was associated with GC in one study [40]. We plan to explore the role of aflatoxins using our stored blood specimens.

Contrary to our expectations, addition of salt at the table was more common in Antofagasta, the low risk area. However, salt intake is a particularly challenging variable for dietary assessment given large intraindividual variation and discretionary use, often not adequately captured [41]. We will explore this further by measuring sodium excretion in urine samples.

We explored gastric atrophy as measured by reductions in the PgI/PgII ratio. Despite limited sensitivity and specificity for disease detection [42], Pgs are strongly associated with prospective GC risk [43] and in the context of this ecologic study we consider them as surrogate markers of gastric cancer risk. In Valdivia, serologic evidence of gastric atrophy was more common than in Antofagasta and occurred at earlier ages, in concordance with the proposed notion that the pattern of gastritis at the population level is an important determinant of differences in GC incidence [27]. However, the explanation of the regional differences in atrophic gastritis (an intermediate outcome) and GC risk is still to be found in bacterial, environmental or genetic factors. In the multivariable model the main risk factors for atrophy were being

from Valdivia, age, chili consumption, a typical element of the diet in Southern Chile, and CagA antibodies. There have been few studies investigating factors associated with gastric atrophy, but older age and CagA seem to be consistently identified [16, 17].

The strengths of our study include recruitment of population-based samples by invitation, preventing selection based on symptoms or attendance to clinics. We used standardized methods for interviewing, specimen collection and laboratory procedures. These methods will also be used in subsequent ENIGMA studies in other countries permitting direct comparison and pooling of results.

Among study limitations, participation rates were lower in Valdivia because of idiosyncratic characteristics plus logistic and climatic reasons. Another limitation is that the difference in GC incidence between the two sites may not be sufficiently large to detect the differences in exposure to risk factors explaining it. We also had small numbers for some analyses and did not have biopsy materials to study the prevalence of histologically confirmed preneoplastic lesions or the *H. pylori* strains present in the two sites. The ELISA assay in this study (Biohit) is a standardized test based on purified *H. pylori* antigens that has been used in multiple populations in various studies. Although the use of local antigens is recommended to assure the validity of serologic methods in clinical practice, in ENIGMA we are using the same test in the different study sites to assure standardization of the results. The fact that the results of the Western Blot test were comparable to those of the ELISA is reassuring in this regard. In addition, this is an ecologic study where the areas are the unit of analysis and the study participants are not the same individuals with gastric cancer in the population, and therefore the associations described should be considered for hypothesis generating and need to be corroborated in individual-based longitudinal studies. The second component of ENIGMA (ENIGMA II) that aims to investigate the worldwide epidemiology of gastric premalignant conditions includes gastroscopy and biopsies in adult participants of the ENIGMA I studies to address these issues.

In conclusion, we found no differences in the prevalence of *H. pylori* or its virulence factors between the high and low risk areas in Chile. *H. pylori* is less common in younger generations and we should expect the trends of declining gastric cancer rates to continue rates in the future. We identified some potential risk factors that could explain both the difference in GC risk and atrophy as its surrogate. However, the role of diet, in particular chili pepper, should be explored further. Our study suggests that atrophy is more frequent and occurs much earlier in the high-risk area of GC and that one of the main risk factors for atrophic gastritis is consumption of chili pepper that could act directly or through interactions with other environmental factors. These hypotheses deserve further research as they could help design additional preventive interventions.

## Supporting information

**S1 Fig. *H. pylori* prevalence determined by ELISA by 10 years age group and gender (both sites combined).**
(PNG)

**S1 Table. Demographic characteristics by study site.**
(DOCX)

**S2 Table. *H. pylori* seropositivity determined by ELISA and Helicoblot and the frequency of *H. pylori* immunoreactive bands by study site among participants aged 40 and older.**
(DOCX)

**S1 File. Interview English.**
(DOCX)

**S2 File. Interview Spanish.**
(DOCX)

**S1 Data. Final dataset.**
(CSV)

**S1 Codebook.**
(XLS)

## Acknowledgments

Dr Paula González, our beloved colleague and leader of cancer research in Latin America passed away before the submission of the final version of this manuscript. Rolando Herrero accepts responsibility for the integrity and validity of the data collected and analyzed.

We thank Drs Sabina Rinaldi and Maria de la Luz Hernandez for their input and support in *H. pylori* ELISA testing at IARC. The members of the ENIGMA Chile study group are the following: Fabio Paredes, Catterina Ferreccio (group leader) (catferre@gmail.com) from the Pontificia Universidad Católica de Chile; Janet Altamirano, Ana Wall, Maritza Corvalán, Felix Díaz, Patricia Ochoa, Francisco Mena, Eliseo Martinez from the Universidad de Antofagasta; José Ignacio Zarate, Juan Carlos Velásquez form the Registro hospitalario de cáncer Hospital Base Valdivia; Solange Vargas from the Registro Poblacional de Cancer Valdivia, SEREMI de Salud de Los Ríos; Ivo Muñoz from the Universidad Austral de Chile; Dace Rudzīte from the University of Latvia; M. Constanza Camargo from the US National Cancer Institute and Salvatore Vaccarella from the International Agency for Research on cancer (IARC).

**Disclaimer:** Where authors are identified as personnel of the International Agency for Research on Cancer / World Health Organization, the authors alone are responsible for the views expressed in this article and they do not necessarily represent the decisions, policy or views of the International Agency for Research on Cancer / World Health Organization.

## Author Contributions

**Conceptualization:** Rolando Herrero, Katy Heise, María Enriqueta Beltrán, Paula Gonzalez, Raul Murillo, Catterina Ferreccio.

**Data curation:** Sylvaine Barbier.

**Investigation:** Johanna Acevedo, Paz Cook, Claudia Gonzalez, Jocelyne Gahona, Raimundo Cortés, Luis Collado, Marcis Leja, Francis Megraud, Maria de la Luz Hernandez, Sylvaine Barbier.

**Methodology:** Katy Heise, Luis Collado, María Enriqueta Beltrán.

**Project administration:** Johanna Acevedo, Paz Cook.

**Resources:** Marcos Cikutovic, Marcis Leja, Francis Megraud.

**Supervision:** Rolando Herrero, Katy Heise, Claudia Gonzalez, Jocelyne Gahona, María Enriqueta Beltrán, Paula Gonzalez, Maria de la Luz Hernandez, Jin Young Park, Catterina Ferreccio.

**Validation:** Rolando Herrero, Marcis Leja, Catterina Ferreccio.

**Writing – original draft:** Rolando Herrero, Katy Heise, Johanna Acevedo, Paz Cook, Paula Gonzalez, Raul Murillo, Jin Young Park, Catterina Ferreccio.

**Writing – review & editing:** Rolando Herrero, Johanna Acevedo, Paz Cook, Claudia Gonza-
lez, Jocelyne Gahona, Raimundo Cortés, Luis Collado, María Enriqueta Beltrán, Marcos
Cikutovic, Paula Gonzalez, Raul Murillo, Marcis Leja, Francis Megraud, Maria de la Luz
Hernandez, Jin Young Park, Catterina Ferreccio.

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
