## [Decision Letter · Decision Letter 0]

19 Feb 2020

PONE-D-19-35057

Regional variations in Helicobacter pylori infection, gastric atrophy and gastric cancer risk: the ENIGMA study in Chile

PLOS ONE

Dear Dr Herrero,

Thank you for submitting your manuscript to PLOS ONE. After careful consideration, we feel that it has merit but does not fully meet PLOS ONE’s publication criteria as it currently stands. Therefore, we invite you to submit a revised version of the manuscript that addresses the points raised during the review process.

Please carefully address all comments raised by both reviewers, paying particular attention to statistical comparisons and the fact that this is an ecologic study and not an analytic study.  The weaknesses of the ecologic study design should be clearly stated in the discussion (e.g., that areas are the unit of analysis, and that the individuals with Hp information are not the same individuals with gastritis or gastric cancer within the study areas).  Ecologic studies may help to define hypotheses that can then be tested using analytic (observational) study designs with individual-level data.  Interventions would only be considered after analytic studies, so the final sentence of the Discussion should be revised accordingly. 

Please review the PLOS ONE data sharing policy and clearly state where the data can be accessed; and if not, the reasons should be specified.  Your revision should also be edited for English grammar. 

We would appreciate receiving your revised manuscript by Apr 04 2020 11:59PM. To enhance the reproducibility of your results, we recommend that if applicable you deposit your laboratory protocols in protocols.io, where a protocol can be assigned its own identifier (DOI) such that it can be cited independently in the future. For instructions see: http://journals.plos.org/plosone/s/submission-guidelines#loc-laboratory-protocols

We look forward to receiving your revised manuscript.

Kind regards,

Eric J. Duell, MS, PhD

Academic Editor

PLOS ONE

Journal Requirements:

3. Thank you for including your ethics statement: The study was approved by IARC Ethics Committee (IEC No.14-17) and local ethical committees (Comité Ética de la Investigación, Escuela de Medicina Pontificia Universidad Católica de Chile, Comité de Ética Zona Norte, Servicio de Salud Coquimbo). Eligible adults signed informed consent. Children between 12 and 18 provided informed assent and parental informed consent, and for children under 12 only parental informed consent was obtained.

4. One of the noted authors is a group or consortium (ENIGMA Chile study group). In addition to naming the author group, please list the individual authors and affiliations within this group in the acknowledgments section of your manuscript. Please also indicate clearly a lead author for this group along with a contact email address.

6. Thank you for stating the following in the Competing Interests section:

We note that one or more of the authors are employed by a commercial company: International Agency for Research on Cancer and Advanced Center for Chronic Diseases.

Reviewers' comments:

Reviewer's Responses to Questions

**Comments to the Author**

1. Is the manuscript technically sound, and do the data support the conclusions?

Reviewer #1: Partly

Reviewer #2: Partly

2. Has the statistical analysis been performed appropriately and rigorously? 

Reviewer #1: No

Reviewer #2: Yes

3. Have the authors made all data underlying the findings in their manuscript fully available?

Reviewer #1: No

Reviewer #2: Yes

4. Is the manuscript presented in an intelligible fashion and written in standard English?

Reviewer #1: No

Reviewer #2: Yes

5. Review Comments to the Author

Reviewer #1: The authors have conducted a survey of H. pylori infection in Chile using conventional methods. Importantly, this prevalence estimates a structured population sample rather than simply controls from a case-control study or other continence sample. Overall the prevalence data on Hp seropositivity and the virulence factors was valuable. The cut points used for the presence of atrophy are OK, but it would be much better to see the actual distributions of PG1, PG2, and the PG ratio in the two communities after standardization for age sex differences in the samples. The atrophy cut points are not optimized for assessing gastric cancer risk, the main purpose of their inclusion here.

Overall, I found much of the presentation extraneous to the stated core goal of the project – the seroprevalence of Hp, CagA, VacA, and Pepsinogen. Drawing conclusions about potential dietary risk factors diminished the focus on what is most valuable about the project.

ABSTRACT

The two sentences of background in the abstract are contradictory. They properly state that the rates of gastric cancer don’t follow prevalence estimates, but then state they do the survey to explain regional differences in gastric cancer risk. The prevalence is important whether it directly explains the rates or not. According to the intro, the rates of GC in Valdivia and Antofagasta (in men) are 33.1 and 21.2, respectively. That isn’t approximately two-fold different as written, although the mortality rate is nearly 3-fold different.

• Please be more specific than ‘virulence factors’ in the methods

• Age standardized to what?

• The comments on dietary factors are supposed to inform the prevalence of Hp, gastric atrophy, or gastric cancer?

• Not clear if they create a single model for assessing risk factors for atrophy across Chile and if they did, does geographic location remain a risk factor?

The first sentence of the conclusion doesn’t seem to be paired to the data presented in the results and it isn’t clear what ‘regional variation’ refers to, gastric cancer or atrophy?

Drawing any conclusions about novel risk factors for gastric cancer, such as chili peppers, from this modest study seems a bit much in the abstract.

The abstract doesn’t present any data on prevalence! I thought that was the primary goal of the study?

MAIN TEXT

H. pylori seropositivity is highly dependent on the senstivitiy of the test. This is less important when making risk estimates comparing case and controls within a population, but here it seems all important given that event6ually they will be comparing these rates to other populations (Page 14 “This is the first of a series of IARC international prevalence surveys.”) An independent assessment of Hp would be really helpful here because there is no guarantee the test will work the same in other parts of the world that have different strains of Hp. Can the authors point to any data on the sensitivity of the Biohit test in Chilean people? It was reassuring that the Eiken and Biohit values are similar.

I am not sure how to interpret the prevalence odds ratios used in figure 3. It seems it would be simpler to just present a table of a figure with the point estimate for each location and then some measure on uncertainty around the prevalence estimate. Or maybe just the net difference in the prevalence? I also don’t know what it means to have an adjusted OR for, example, of chili intake. Chili consumption if ~5-fold more common in Valdivia (24%) than in Antofagasta (5%), so what does the OR of 8.6 mean and in what sense did the other factors in table 3 confound the chili intake estimate?

The prevalence of atrophy seems to parallel the incidence of gastric cancer quite nicely for both the lower and more stringent cut point. It would be nice if the authors showed us all values (and ratios ) for the two areas for GC incidence, GC mortality, Hp positvity, CagA, VacA, Pep1, Pep2, and PG ratio, and atrophy in a single table

Page 14, line 302 – please don’t use the term ‘developed countries’ with out a modifier such as economically-developed countries.

I don’t understand this conclusion on page 18 drawn from the analysis presented in figure 5. “…one of the main risk factors for atrophic gastritis is consumption of chili pepper.” An OR of 1.97 and a prevalence of 22% in cases seems like a modest association in a single study. In fact, chili pepper consumption would have no individual level predictive value for whether or not an individual has atrophy. In fact, lack of education has a stronger association (OR of almost 3) and the same prevalence. Why was CagA included rather than the whole cell Hp antibody? I don’t see any evidence that CagA was more predictive that just seropositivity. In some previous studies, disease risk is greatest in those that are CagA positive and whole cell negative, but it doesn’t seem that was explored here.

Finally, an established risk factor for gastric cancer, tobacco seems to be missing from the results. The methods noted that smoking was collected so it should be included in the report.

Reviewer #2: This is an ecologic study of Helicobacter pylori, gastric atrophy, selected aspects of diet and gastric from two areas of Chile with different gastric cancer mortality.

The only consistent finding is higher atrhopy in the high risk area. The others, with the exception of vegetable consumption, are difficult to explain.

Given the limitations of the study design, their interpretation therefore, should be more cautious.

6. PLOS authors have the option to publish the peer review history of their article (what does this mean?). If published, this will include your full peer review and any attached files.

Reviewer #1: No

Reviewer #2: Yes: Carlo La Vecchia

---

## [Author Response · Author response to Decision Letter 0]

27 Jun 2020

7 April 2020

Eric J Duell 

Academic Editor

PLOS ONE

PONE-D-19-35057

Regional variations in Helicobacter pylori infection, gastric atrophy and gastric cancer risk: the ENIGMA study in Chile

Dear Dr Duell:

Thanks for the careful review of our manuscript and request for revisions. Please find below our responses to the Editor and reviewers’ comments

Editor’s comment:

The weaknesses of the ecologic study design should be clearly stated in the discussion (e.g., that areas are the unit of analysis, and that the individuals with Hp information are not the same individuals with gastritis or gastric cancer within the study areas). Ecologic studies may help to define hypotheses that can then be tested using analytic (observational) study designs with individual-level data. Interventions would only be considered after analytic studies, so the final sentence of the Discussion should be revised accordingly. 

The Editor’s suggestion has been considered and the following text has been added to the discussion: ‘In addition, this is an ecologic study where the areas are the unit of analysis and the study participants are not the same individuals with gastric cancer in the population. Therefore, the associations described should be considered hypothesis-generating and need to be corroborated in individual-based longitudinal studies.’

Journal requirements

The manuscript and tables have been prepared following PLOS ONE’s style requirements, including those for file naming

We have included more details in the methods section about the questionnaire and submitted the actual questionnaire in English and Spanish as supporting documentation

3. Thank you for including your ethics statement: The study was approved by IARC Ethics Committee (IEC No.14-17) and local ethical committees (Comité Ética de la Investigación, Escuela de Medicina Pontificia Universidad Católica de Chile, Comité de Ética Zona Norte, Servicio de Salud Coquimbo). Eligible adults signed informed consent. Children between 12 and 18 provided informed assent and parental informed consent, and for children under 12 only parental informed consent was obtained. Please amend your current ethics statement to include the full name of the ethics committee/institutional review board(s) that approved your specific study.

The ethics statement has been modified to include the full name of the IRBs that approved our study. 

Once you heave amended this/these statement(s) in the Methods section of the manuscript, please add the same text to the “Ethics Statement” field of the submission form (via “Edit Submission”).

The modified text has been added to the Ethics Statement

4. One of the noted authors is a group or consortium (ENIGMA Chile study group). In addition to naming the author group, please list the individual authors and affiliations within this group in the acknowledgments section of your manuscript. Please also indicate clearly a lead author for this group along with a contact email address.

The authors and affiliations have been moved to the acknowledgements section and the team leader and her email are now indicated

We have removed the reference to Data not shown, it is not necessary.

6. Thank you for stating the following in the Competing Interests section: "The authors have declared that no competing interests exist." We note that one or more of the authors are employed by a commercial company: International Agency for Research on Cancer and Advanced Center for Chronic Diseases.

None of these organizations are commercial companies. IARC is a research agency of the World Health Organization and as such it is a United Nations institution. The Advanced Center for Chronic Diseases, ACCDiS, is a research center publicly funded by the Chilean Government, after a national competition, the Direction is shared by the two main research Universities in Chile: University of Chile, and Pontificia Universidad Católica de Chile, both public non-for-profit organizations.

There is no commercial affiliation as described above

b. Please also provide an updated Competing Interests Statement declaring this commercial affiliation along with any other relevant declarations relating to employment, consultancy, patents, products in development, or marketed products, etc. Within your Competing Interests Statement, please confirm that this commercial affiliation does not alter your adherence to all PLOS ONE policies on sharing data and materials by including the following statement: "This does not alter our adherence to PLOS ONE policies on sharing data and materials.” (as detailed online in our guide for authors http://journals.plos.org/plosone/s/competing-interests) . If this adherence statement is not accurate and there are restrictions on sharing of data and/or materials, please state these. Please note that we cannot proceed with consideration of your article until this information has been declared. Please include both an updated Funding Statement and Competing Interests Statement in your cover letter. We will change the online submission form on your behalf.

As described in a. there are no commercial organizations involved

 

Reviewers' comments:

……

Review Comments to the Author

Reviewer #1: The authors have conducted a survey of H. pylori infection in Chile using conventional methods. Importantly, this prevalence estimates a structured population sample rather than simply controls from a case-control study or other continence sample. Overall the prevalence data on Hp seropositivity and the virulence factors was valuable. 

We appreciate the positive comment of the reviewer

The cut points used for the presence of atrophy are OK, but it would be much better to see the actual distributions of PG1, PG2, and the PG ratio in the two communities after standardization for age sex differences in the samples.

We have included in the text a description of the mean values of the PG1/PG2 ratios, indicating that there were no differences by sex (table below for the reviewers)

 Antofagasta Valdivia 

 N=297 N=308 

Pepsinogen mean(SD) mean(SD) P values1

PGI 71.0 (44.5) 67.4 (43.2) 0.318

PGII 20.1 (13.0) 21.2 (13.3) 0.321

Ratio 4.0 (1.8) 3.6 (1.8) 0.008

1 Student t-test

 Women Men 

 N=354 N=264 

Pepsinogen mean(SD) mean(SD) P values1

PGI 71 (46.7) 66.6 (39.5) 0.2164

PGII 21.3 (14.5) 19.7 (10.9) 0.1193

Ratio 3.8 (1.9) 3.7 (1.7) 0.519

1 Student t-test with unequal variances 

2 Student t-test with equal variances 

The atrophy cut points are not optimized for assessing gastric cancer risk, the main purpose of their inclusion here.

We agree with the reviewer that the atrophy cutpoints are not aimed at directly estimating gastric cancer risk. Instead, they are generally accepted cutoffs to indicate the presence of gastric atrophy, which in this context represents a surrogate marker of gastric cancer risk, given the known association of atrophy with risk of gastric cancer. Additional text has been included in the discussion to clarify this aspect, as follows: ‘We explored gastric atrophy as measured by reductions in the Pg1/Pg2 ratio. Despite limited sensitivity and specificity for disease detection [42], Pgs are strongly associated with prospective GC risk [43] and in the context of this ecologic study we consider them as surrogate markers of gastric cancer risk.’

Overall, I found much of the presentation extraneous to the stated core goal of the project – the seroprevalence of Hp, CagA, VacA, and Pepsinogen. Drawing conclusions about potential dietary risk factors diminished the focus on what is most valuable about the project.

We accept the suggestion of the reviewer. Throughout the paper and following the other comments and the observation of the other reviewer, we are now emphasizing more the findings about prevalence and the differences in atrophy between the two centers than the dietary associations.

ABSTRACT

The two sentences of background in the abstract are contradictory. They properly state that the rates of gastric cancer don’t follow prevalence estimates, but then state they do the survey to explain regional differences in gastric cancer risk. The prevalence is important whether it directly explains the rates or not.

We accept the critique and modified the description of the aims as follows: ‘The IARC worldwide H. pylori prevalence surveys (ENIGMA) aim at systematically describing age and sex-specific prevalence of H. pylori infection around the world and generating hypotheses to explain regional variations in gastric cancer risk’. We think this removes the contradiction described by the reviewer.

According to the intro, the rates of GC in Valdivia and Antofagasta (in men) are 33.1 and 21.2, respectively. That isn’t approximately two-fold different as written, although the mortality rate is nearly 3-fold different.

This has been modified to avoid the inconsistency and now reads: We selected age- and sex-stratified population samples in two areas with different gastric cancer incidence and mortality in Chile.

Please be more specific than ‘virulence factors’ in the methods

We have modified the sentence and now it mentions the specific virulence factors tested for.

 Age standardized to what?

To the world population, as described in the statistical methods. Modified in the abstract for clarity.

The comments on dietary factors are supposed to inform the prevalence of Hp, gastric atrophy, or gastric cancer?

It is meant to inform gastric cancer differences in the context of the ecological analysis. It was modified to read: ‘The comparison of the prevalence of known and potential cofactors of H pylori in gastric carcinogenesis between the high and the low risk area showed that consumption of chili products ….’ There is also a new section in the discussion where we discuss the ecologic nature of the study as a limitation, as proposed by the Editor.

Not clear if they create a single model for assessing risk factors for atrophy across Chile and if they did, does geographic location remain a risk factor?

Correct, we created a single model to assess risk factors for atrophy across Chile (figure 5), and geographic location remained a risk factor associated with an OR of 1.67 (95%CI=1.1-2.5). A clarification was added to the abstract.

The first sentence of the conclusion doesn’t seem to be paired to the data presented in the results and it isn’t clear what ‘regional variation’ refers to, gastric cancer or atrophy?’

This sentence was modified and does not use the phrase ‘’regional variation’’ anymore and is now more consistent with the data presented. Now it reads: ‘The prevalence of H. pylori infection and its virulence factors was similar in the high and the low risk area, but atrophy was more common and occurred at younger ages in the higher risk area’.

Drawing any conclusions about novel risk factors for gastric cancer, such as chili peppers, from this modest study seems a bit much in the abstract.

We modified the conclusion in the abstract to say just ‘dietary factors’, reducing the emphasis on chili pepper’. There is also additional paragraph in the discussion about the interpretation of the findings in an ecologic study.

The abstract doesn’t present any data on prevalence! I thought that was the primary goal of the study?

We now include more details about the prevalence of H pylori in the abstract as follows: ’H. pylori seroprevalence (age-standardized to world population) and antibodies against CagA and VacA were similar in both sites. H. pylori seroprevalence was 20% among children <10 years old, 40% among 10-19 year olds, 60% in the 20-29 year olds and close to or above 80% in those 30+ years’.

H. pylori seropositivity is highly dependent on the sensitivity of the test. This is less important when making risk estimates comparing case and controls within a population, but here it seems all important given that event6ually they will be comparing these rates to other populations (Page 14 “This is the first of a series of IARC international prevalence surveys.”) An independent assessment of Hp would be really helpful here because there is no guarantee the test will work the same in other parts of the world that have different strains of Hp. Can the authors point to any data on the sensitivity of the Biohit test in Chilean people? It was reassuring that the Eiken and Biohit values are similar.

Thanks for this important comment given the aim of the ENIGMA study is to compare seroprevalence between countries. We consider that using the same methodology offers advantages for standardization of results from country to country making them comparable. However, validation of serologic assays with local antigens has been proposed for use of serology in the clinical context. In our view, this could introduce additional variability and validation of the method in each country is at this point out of the scope of the ENIGMA studies. The ELISA method we used is based on multiple antigens from the bacterium and has been used in many epidemiologic studies in various populations. In addition, the concordance of our results between the ELISA and the Western Blot, which use antigens of different sources are reassuring as indicated by the reviewer. Additional text has been added in the discussion as part of the potential limitations of our study as follows: ‘The ELISA assay in this study (Biohit) is a standardized test based on purified H. pylori antigens that has been used in multiple studies. Although the use of local antigens is recommended to assure the validity of serologic methods in clinical practice, in ENIGMA we are using the same test in the different study sites to assure standardization. The fact that the results of the Western Blot test were comparable to those of the ELISA is reassuring in this regard.’

I am not sure how to interpret the prevalence odds ratios used in figure 3. It seems it would be simpler to just present a table of a figure with the point estimate for each location and then some measure on uncertainty around the prevalence estimate. Or maybe just the net difference in the prevalence? 

Figure 3 is an effort to summarize the information: the first two columns present the prevalence for each site, the right part presents the adjusted odds ratio for that variable in the high-risk vs low risk site. We clarify this now in the footnote of the figure 3.

I also don’t know what it means to have an adjusted OR for, example, of chili intake. Chili consumption if ~5-fold more common in Valdivia (24%) than in Antofagasta (5%), so what does the OR of 8.6 mean and in what sense did the other factors in table 3 confound the chili intake estimate?

All significant factors were included in the logistic model, and we retained those significantly associated to the high-risk area. The adjusted OR was not similar to the unadjusted prevalence ratio because there were several variables that when included in the model made the association of chili pepper to Valdivia more evident and improved the fit. The OR of 8.6 indicated that if we take into account the effect of other variables and their correlations, the odds of consuming chili are much higher if you are from Valdivia. We added in the discussion that chili consumption is a typical element of the diet in Sothern CHile

The prevalence of atrophy seems to parallel the incidence of gastric cancer quite nicely for both the lower and more stringent cut point. It would be nice if the authors showed us all values (and ratios ) for the two areas for GC incidence, GC mortality, Hp positvity, CagA, VacA, Pep1, Pep2, and PG ratio, and atrophy in a single table

Thanks for the suggestion. The information about each indicator is presented in the different sections of the article following an order that we consider logical based on the importance of the markers. Including another table with the summarized information would be repetitive. On the other hand, removing the other tables where the information is now presented and using only one table is a possibility but that would somehow alter the flow of the text. We prefer to keep it the way it is now but we are totally open to changing based on the reviewer and editor’s recommendation.

Page 14, line 302 – please don’t use the term ‘developed countries’ without a modifier such as economically-developed countries.

The sentence was modified and the term was replaced by high-income countries.

I don’t understand this conclusion on page 18 drawn from the analysis presented in figure 5. “…one of the main risk factors for atrophic gastritis is consumption of chili pepper.” An OR of 1.97 and a prevalence of 22% in cases seems like a modest association in a single study. In fact, chili pepper consumption would have no individual level predictive value for whether or not an individual has atrophy. In fact, lack of education has a stronger association (OR of almost 3) and the same prevalence. 

We have modified the discussion to include more details about the different factors associated with atrophy and de-emphasized the chili pepper finding as a novel association that requires further exploration.

Why was CagA included rather than the whole cell Hp antibody? I don’t see any evidence that CagA was more predictive that just seropositivity. In some previous studies, disease risk is greatest in those that are CagA positive and whole cell negative, but it doesn’t seem that was explored here.

The whole cell Hp antibody was included in the model but it was not associated with risk of atrophy, while CagA was associated. We now explain this in more detail in the results section.

Finally, an established risk factor for gastric cancer, tobacco seems to be missing from the results. The methods noted that smoking was collected so it should be included in the report.

In Chile, tobacco is expensive and low-income people (who are a high-risk group for GC) smoke less than higher income people, thus smoking is always confounded by age, sex and socioeconomic status. Current smoking was more common in Valdivia (34% vs 25%), but the association was not significant in the adjusted model. It was also not associated with atrophy in the adjusted model. We have added details of these findings in the text

 Reviewer #2: This is an ecologic study of Helicobacter pylori, gastric atrophy, selected aspects of diet and gastric from two areas of Chile with different gastric cancer mortality.

The only consistent finding is higher atrophy in the high risk area. The others, with the exception of vegetable consumption, are difficult to explain.

Given the limitations of the study design, their interpretation therefore, should be more cautious.

Based on this comment and those of the other reviewer, we are now giving more importance to the finding in relation to atrophy than the dietary factors and have added several caveats about the interpretation of the findings in an ecologic study.

---

## [Decision Letter · Decision Letter 1]

29 Jul 2020

Regional variations in Helicobacter pylori infection, gastric atrophy and gastric cancer risk: the ENIGMA study in Chile

PONE-D-19-35057R1

Dear Dr. Herrero,

We’re pleased to inform you that your manuscript has been judged scientifically suitable for publication and will be formally accepted for publication once it meets all outstanding technical requirements.

Kind regards,

Eric J. Duell, MS, PhD

Academic Editor

PLOS ONE

Additional Editor Comments (optional):

All reviewer comments have been addressed.

Reviewers' comments:

Reviewer's Responses to Questions

**Comments to the Author**

1. If the authors have adequately addressed your comments raised in a previous round of review and you feel that this manuscript is now acceptable for publication, you may indicate that here to bypass the “Comments to the Author” section, enter your conflict of interest statement in the “Confidential to Editor” section, and submit your "Accept" recommendation.

Reviewer #1: All comments have been addressed

Reviewer #2: All comments have been addressed

2. Is the manuscript technically sound, and do the data support the conclusions?

Reviewer #1: Yes

Reviewer #2: Yes

3. Has the statistical analysis been performed appropriately and rigorously? 

Reviewer #1: Yes

Reviewer #2: Yes

4. Have the authors made all data underlying the findings in their manuscript fully available?

Reviewer #1: Yes

Reviewer #2: Yes

5. Is the manuscript presented in an intelligible fashion and written in standard English?

Reviewer #1: Yes

Reviewer #2: Yes

6. Review Comments to the Author

Reviewer #1: (No Response)

Reviewer #2: None additional.

None additional.

None additional.

None additional.

None additional.

None additional.

7. PLOS authors have the option to publish the peer review history of their article (what does this mean?). If published, this will include your full peer review and any attached files.

Reviewer #1: No

Reviewer #2: **Yes: **Carlo La Vecchia

---

## [Editor Report · Acceptance letter]

25 Aug 2020

PONE-D-19-35057R1 

Regional variations in Helicobacter pylori infection, gastric atrophy and gastric cancer risk: the ENIGMA study in Chile 

Dear Dr. Herrero:

I'm pleased to inform you that your manuscript has been deemed suitable for publication in PLOS ONE. Congratulations! Your manuscript is now with our production department. 

Kind regards, 

on behalf of

Dr. Eric J. Duell 

Academic Editor

PLOS ONE